# LEARNING FEATURES WITH PARAMETER-FREE LAYERS

**Dongyoon Han**[1]**, YoungJoon Yoo**[1,2]**, Beomyoung Kim**[2]**, Byeongho Heo**[1]
[1]NAVER AI Lab, [2]NAVER CLOVA

## ABSTRACT

Trainable layers such as convolutional building blocks are the standard network design choices by learning parameters to capture the global context through successive spatial operations. When designing an efficient network, trainable layers such as the depthwise convolution is the source of efficiency in the number of parameters and FLOPs, but there was little improvement to the model speed in practice. This paper argues that simple built-in parameter-free operations can be a favorable alternative to the efficient trainable layers replacing spatial operations in a network architecture. We aim to break the stereotype of organizing the spatial operations of building blocks into trainable layers. Extensive experimental analyses based on layer-level studies with fully-trained models and neural architecture searches are provided to investigate whether parameter-free operations such as the max-pool are functional. The studies eventually give us a simple yet effective idea for redesigning network architectures, where the parameter-free operations are heavily used as the main building block without sacrificing the model accuracy as much. Experimental results on the ImageNet dataset demonstrate that the network architectures with parameter-free operations could enjoy the advantages of further efficiency in terms of model speed, the number of the parameters, and FLOPs. Code and ImageNet pretrained models are available at https://github.com/naver-ai/PfLayer.

## 1 INTRODUCTION

Image classification has been advanced with deep convolutional neural networks (Simonyan & Zisserman, 2015; Huang et al., 2017; He et al., 2016b) with the common design paradigm of the network building blocks with trainable spatial convolutions inside. Such trainable layers with learnable parameters effectively grasp attentive signals to distinguish input but are computationally heavy. Rather than applying pruning or distillation techniques to reduce the computational cost, developing new efficient operations has been another underlying strategy. For example, a variant of the regular convolution, such as depthwise convolution (Howard et al., 2017) has been proposed to bring more efficiency by reducing the inter-channel computations. The operation has benefits in the computational budgets, including the number of parameters and FLOPs. However, the networks heavily using the depthwise convolution (Howard et al., 2017; Sandler et al., 2018; Tan & Le, 2019) have an inherent downside of the latency, which generally do not reach the speed of the regular convolution.

In a line of study of efficient operations, there have been many works (Wang et al., 2018; Wu et al., 2018; Han et al., 2020; Tan & Le, 2021) based on the regular convolution and the depthwise convolution. Most methods utilize the depthwise convolution's efficiency or target FLOPs-efficiency but are slow in the computation. Meanwhile, parameter-free operations were proposed; a representative work is the Shift operation (Wu et al., 2018). Its efficiency stems from the novel operation without learning spatial parameters by letting the feature mixing convolutions learn from the shifted features. However, the implementation does not bring about the actual speed-up as expected. This is because the operation-level optimization is still demanding compared to the regular convolution with highly optimized performance. Another parameter-free operation feature shuffling (Zhang et al., 2018) is a seminal operation that reduces the computational cost of the feature mixing layer. However, it hardly plays the role of a spatial operation.

In this paper, we focus on efficient parameter-free operations that actually replace trainable layers for network design. We revisit the popular parameter-free operations, the max-pool and the avg-pool op-

erations (*i.e.*, layers), which are used in many deep neural networks (Simonyan & Zisserman, 2015; Huang et al., 2017) restricted to perform downsampling (Simonyan & Zisserman, 2015; Howard et al., 2017; Sandler et al., 2018; He et al., 2016b). Can those simple parameter-free operations be used as the main network building block? If so, one can reduce a large portion of the parameters and the overall computational budget required during training and inference. To answer the question, the max-pool and the avg-pool operations are chosen to be demonstrated as representative simple parameter-free operations. We first conduct comprehensive studies on the layer replacements of the regular convolutions inside networks searched upon the baseline models with model training. Additionally, we incorporate a neural architecture search (Liu et al., 2019) to explore effective architectures based on the operation list with the parameter-free operations and convolutional operations. Based on the investigations, we provide a simple rule of thumb to design an efficient architecture using the parameter-free operations upon primitive network architectures. The design guide is applied to popular heavy networks and validated by the performance trained on ImageNet (Russakovsky et al., 2015). It turns out that our models have apparent benefits to the computational costs, particularly the faster model speeds. In addition, ImageNet-C (Hendrycks & Dietterich, 2019) and ImageNet-O (Hendrycks et al., 2021) results show our models are less prone to be overfitting. We further propose a novel deformable parameter-free operation based on the max-pool and the avg-pool to demonstrate the future of a parameter-free operation. Finally, we show that the parameter-free operations can successfully replace the self-attention layer (Vaswani et al., 2017), thus attaining further efficiency and speed up. We summarize our contributions as follows:

- We study whether parameter-free operations can replace trainable layers as a network building block. To our knowledge, this is the first attempt to investigate a simple, built-in parameter-free layer as a building block for further efficiency (§3).

- We provide a rule of thumb for designing a deep neural network including convolutional neural networks and vision transformers with parameter-free layers (§4).

- Experimental results show that our efficient models outperform the previous efficient models and yield faster model speeds with further robustness (§5).

## 2 PRELIMINARIES

A network building block where trainable layers are inside is a fundamental element of modularized networks (Xie et al., 2017; Sandler et al., 2018; Tan & Le, 2019). We start a discussion with the elementary building block and then explore more efficient ones.

### 2.1 BASIC BUILDING BLOCKS

**Convolution Layer.** We recall the regular convolutional operation by formulating with matrix multiplication first. Let $f \in \mathcal{R}^{c_{in} \times H \times W}$ as the input feature, the regular convolution of the feature $f$ with kernel size $k$ and stride $r$ is given as

$$y_{o,i,j} = \sigma \left( \sum_{h,w=-\lfloor k/2 \rfloor}^{\lfloor k/2 \rfloor} \sum_{u=1}^{c_{in}} W_{o,u,h,w} \cdot f_{u,r*i+h,r*j+w} \right), \tag{1}$$

where $W$ denotes the weight matrix, and the function $\sigma(\cdot)$ denotes an activation function such as ReLU (Nair & Hinton, 2010) with or without batch normalization (BN) (Ioffe & Szegedy, 2015). The convolution itself has been a building block due to the expressiveness and design flexibility. VGG (Simonyan & Zisserman, 2015) network was designed by accumulating the 3×3 convolution to substitute the convolutions with larger kernel sizes but still have high computational costs.

**Bottleneck Block.** We now recall the bottleneck block, which primarily aimed at efficiency. We represent the bottleneck block by the matrix multiplication as equation 1:

$$y_{o,i,j} = \sigma \left( \sum_{v=1}^{\rho c_{in}} P_{o,v} \cdot \sigma \left( \sum_{h,w=-\lfloor k/2 \rfloor}^{\lfloor k/2 \rfloor} \sum_{u=1}^{\rho c_{in}} W_{v,u,h,w} \cdot g_{v,r*i+h,r*j+w} \right) \right), \tag{2}$$

where $g_{o,i,j} = \sigma(\sum_{u=1}^{c_{in}} Q_{o,u} \cdot f_{u,i,j})$, and the matrix $P$ and $Q$ denote the weights of 1×1 convolutions with the inner dimensions $\rho c_{in}$. This design regime is efficient in terms of the computational budgets and even proven to be effective in the generalization ability when stacking up the bottleneck blocks compared with the basic blocks. Albeit a 3×3 convolution is replaced with two feature

mixing layers (*i.e.*, 1×1 convolutions), the expressiveness is still high enough with a low channel expansion ratio $\rho$ (*i.e.*, $\rho = 1/4$ in ResNet (He et al., 2016b; Xie et al., 2017)). However, due to the presence of the regular 3×3 convolution, only adjusting $\rho$ hardly achieves further efficiency.

## 2.2 EFFICIENT BUILDING BLOCKS

**Inverted Bottleneck.** The grouped operations, including the group convolution (Xie et al., 2017) and the depthwise convolution (Howard et al., 2017)) have emerged as more efficient building blocks. Using the depthwise convolution inside a bottleneck (Sandler et al., 2018) is called the inverted bottleneck, which is represented as

$$y_{o,i,j} = \sigma \left( \sum_{v=1}^{\rho c_{in}} P_{o,v} \cdot \sigma \left( \sum_{h,w=-\lfloor k/2 \rfloor}^{\lfloor k/2 \rfloor} W_{v,h,w} \cdot g_{v,r*i+h,r*j+w} \right) \right), \tag{3}$$

where the summation over the channels in equation 2 has vanished. This slight modification takes advantage of further generalization ability, and therefore stacking of the blocks leads to outperform ResNet with a better trade-off between accuracy and efficiency as proven in many architectures (Tan & Le, 2019; Han et al., 2021). The feature mixing operation in equation 2 with the following point-wise activation function (*i.e.*, ReLU) may offer a sufficient rank of the feature with the efficient operation. However, the inverted bottleneck and the variants below usually need a large expansion ratio $\rho > 1$ to secure the expressiveness (Sandler et al., 2018; Howard et al., 2019; Tan & Le, 2019), so the actual speed is hampered by the grouped operation that requires more optimization on GPU (Gibson et al., 2020; Lu et al., 2021).

**Variants of Inverted Bottleneck.** More refinements on the bottleneck block could bring further efficiency; the prior works (Wang et al., 2018; Han et al., 2020; Tan & Le, 2021; Wu et al., 2018) fully or partially redesigned the layers in the bottleneck with new operations, and therefore theoretical computational costs have been decreased. VersatileNet (Wang et al., 2018) replaced the convolutions with the proposed filters consisting of multiple convolutional filters; GhostNet (Han et al., 2020) similarly replaced the layers with the proposed module that concatenates a preceding regular convolution and additional depthwise convolution. Rather than involving new operations, a simple replacement by simplifying the operations has been proposed instead. EfficientNetV2 (Tan & Le, 2021) improved the training speed by fusing the pointwise and the depthwise convolutions to a single regular convolution. This takes advantage of using the highly-optimized operation rather than using new unoptimized operations. ShiftNet (Wu et al., 2018) simplify the depthwise convolution to the shift operation, where the formulation is very similar to equation 3:

$$y_{o,i,j} = \sigma \left( \sum_{v=1}^{\rho c_{in}} P_{o,v} \cdot \sigma \left( \sum_{h,w=-\lfloor k/2 \rfloor}^{\lfloor k/2 \rfloor} W_{v,h,w} \cdot g_{v,r*i+h,r*j+w} \right) \right), \tag{4}$$

where $W_{v,:,:}$ is simplified to have one 1 and 0 for the rest all $h, w$ for each $v$. This change involves a new operation, so-called Shift, that looks like shifting the features after the computation, and the preceding and the following pointwise convolutions mix the shifted output. The shift operation is actually classified as a parameter-free operation, but a large expansion ratio is yet needed to hold the expressiveness. Furthermore, the implementation is not as readily optimized as regular operations, even in CUDA implementation (He et al., 2019; Chen et al., 2019a).

## 3 EFFICIENT BUILDING BLOCK WITH PARAMETER-FREE OPERATIONS

In this section, we extend an efficient building block by incorporating parameter-free operations. We empirically study whether the parameter-free operations can be used as the main building block in deep neural networks like regular convolution.

### 3.1 MOTIVATION

The learning mechanism of ResNets (He et al., 2016b;a) has been analyzed as the iterative unrolled estimation (Greff et al., 2017; Jastrzebski et al., 2018) which iteratively refine the features owing to the presence of skip connections. This indicates that some layers do not contribute much to learning, which has also been revealed in layer-drop experiments (Veit et al., 2016; Huang et al., 2016). In this light, we argue some building blocks in a residual network can be replaced with parameter-free operations. We investigate the replacement of the spatial operation in popular networks with parameter-free operations to rethink the common practice of network design.

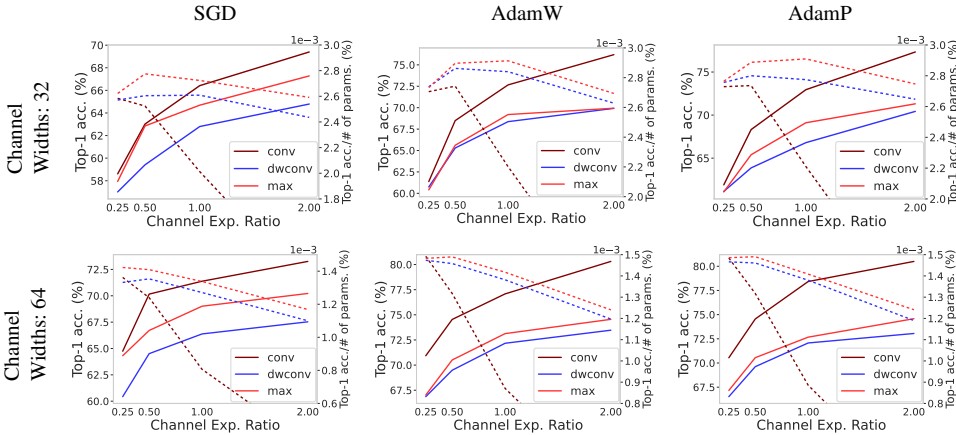

Figure 1. **Single bottleneck study.** We visualize top-1 accuracy (**solid lines**) trained with the different setups including 1) varying channel expansion ratios inside a bottleneck; 2) the different channel widths: 32 (**upper row**) and 64 (**lower row**); 3) diverse optimizers: SGD (**left**), AdamW (**middle**), and AdamP (**right**); We further plot accuracy per # parameters (**dashed lines**) to show the parameter-efficiency of the operations. We observe the regular convolutions work well but be replaceable at a low channel expansion ratio; the alternative operations are highly efficient; the max-pool consistently beats the depthwise convolution.

## 3.2 RETHINKING PARAMETER-FREE OPERATIONS

Our goal is to design an efficient building block that boosts model speed in practice while maintaining model accuracy. Based on equation 2, we remove the inter-channel operation similarly done in equation 3 and equation 4. Then, instead of assigning $W_{v,h,w}$ to be a single value, as in equation 4, we let $W$ have a dependency on the feature $g$ (*i.e.*, $W_{v,h,w} = s(g_{v,r*i+h,r*j+w})$) by introducing a function $s(\cdot)$. Then, the layer would have different learning dynamics interacting with $P$ and $Q$.

Among many candidates to formalize $s(\cdot)$, we allocate the function with a simple one that does not have trainable parameters. We let $s(\cdot)$ pick some large values of $g$ in the range of all $h, w$ per each $v$ like the impulse function. Here, we simplify the function $s(\cdot)$ again to use only the largest values of $g$, where we have the representation of $W_{v,h^*,w^*} = 1$, $(h^*, w^*) = \text{argmax}_{(h,w)} g_{v,r*i+h,r*j+w}$ and other $W_{v,h,w}$ to be 0 in equation 3. One may come up with another simple option: let $s(\cdot)$ compute the moments such as mean or variance of $g$. In practice, those operations can be replaced with built-in spatial pooling operations, which are highly optimized at the operation level. In the following section, we empirically study the effectivness of efficient parameter-free operations in network designs.

## 3.3 EMPIRICAL STUDIES

**On a Single Bottleneck.**   Our study begins with a single bottleneck block of ResNet (He et al., 2016b). We identify such a parameter-free operation can replace the depthwise convolution which is a strong alternative of the regular convolution. We train a large number of different architectures on CIFAR10 and observe the accuracy trend of trained models. The models consist of a single bottleneck, but the channel expansion ratio inside a bottleneck varies from 0.25 to 2, and there are two options for the base-channel width (32 and 64). Training is done with three different optimizers of SGD, AdamW (Loshchilov & Hutter, 2017b), and AdamP (Heo et al., 2021a). Finally, we have total $4 \times 2 \times 3 \times 3 = 72$ models for the study. Fig.1 exhibit that the parameter-free operation consistently outperforms the depthwise convolution in a bottleneck[1]. The regular convolutions clearly achieved higher absolute accuracy, but the efficiency (the accuracy over the number of parameters) is getting worse for larger expansion ratios. We observe that the accuracy gaps between the regular convolution and the parameter-free operation are low for small channel expansion ratios (especially at the ratio=1/4 where the original bottleneck (He et al., 2016b; Xie et al., 2017) adopts). It means that the regular convolution in the original bottleneck block with the expansion ratio of 1/4 can be replaced by the parameter-free operation.

---

[1]The bottleneck with the max-pool operation here is designed by following the formulation in §3.2; we call it the efficient bottleneck. Note that the max-pool operation needs to be adopted due to the performance failures of using the avg-pool operation (see Appendix B).

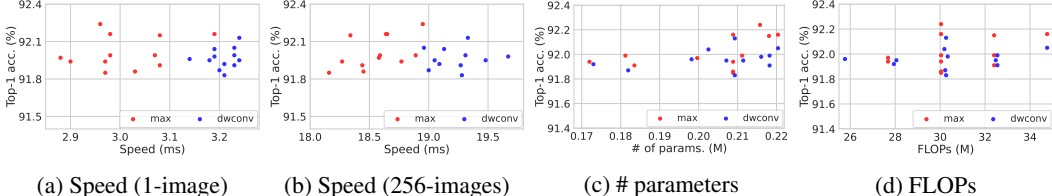

(a) Speed (1-image)  (b) Speed (256-images)  (c) # parameters  (d) FLOPs

Figure 2. **Multiple bottlenecks study**. We respectively pick the 20% best-performing models and visualize in the comparison graphs: (a) Accuracy vs. speed with the batch size of 1; (b) Accuracy vs. speed with the batch size of 256; (c) Accuracy vs. # parameters; (d) Accuracy vs. FLOPs (use FLOPs to mean multiply-adds). The depthwise convolution (blue dots) is an efficient operation in # parameters and FLOPs, but the parameter-free operation (red dots) has a clear benefit in the model speed in practice. A particular parameter-free operation can be an alternative to the depthwise convolution when replacing the regular convolutions.

**On Multiple Bottlenecks.**  We extend the single bottleneck study to deeper networks having multiple bottleneck blocks. We similarly study the bottleneck with different spatial operations, including the depthwise convolution and parameter-free operations, replacing the regular convolution. We choose the max-pool operation again as the parameter-free operation and compare it with the depthwise convolution. To study the impact of repeatedly using parameter-free operations in-depth, we exhaustively perform the layer replacements at layer-level upon the baseline network. We report the trade-off between accuracy and speed by replacing the regular convolutions with the max-pool operation and the depthwise convolution. We use the baseline network, which has eight bottleneck blocks in a standard ResNet architecture (i.e., ResNet-26), and the study has done by replacing each block have efficient operations (i.e., the efficient bottleneck and the bottleneck with the depthwise convolution). We train all the networks with large epochs (300 epochs) due to the depthwise convolution. Fig.2 illustrates the trade-offs of top-1 accuracy and the actual model speed; we observe the models with the parameter-free operations are faster than those with the depthwise convolution as the model accuracies are comparable.

Table 1. **Neural architecture search with individual cell searches.** We investigate how parameter-free operations are chosen with convolutional operations. Six different normal cells are searched individually, and two reduction cells are searched by a unified single cell. Surprisingly, the parameter-free operations are consistently found regardless of the settings. The parameter-free operations in each cell are represented as max @[($d$, $x_1$;...; $x_i$)] and avg @[($d$, $x_1$;...; $x_i$)] which denote the max-pool_3×3 and the avg-pool_3×3 appear at every edge towards $x_1 \ldots x_i$-th nodes in $d$-th (from the input) cell, respectively.

| # of Nodes | Seed No. | Normal Cells | Reduction Cell | Prec-1 |
|---|---|---|---|---|
| 4 (unified cells) | 1 | no parameter-free ops. | avg @[(3, 2), (6, 2)] | 87.84% |
| 1 | 1 | max @[(5, 1)] | max @[(3, 1), (6, 1)] | 86.15% |
| 1 | 2 | max @[(2, 1)] | max @[(3, 1), (6, 1)] | 86.27% |
| 1 | 3 | max @[(2, 1)] | max @[(3, 1), (6, 1)] | 85.42% |
| 2 | 1 | max @[(2, 2), (5, 1;2)] | no parameter-free ops. | 87.64% |
| 2 | 2 | max @[(4, 2), (5, 1;2), (7, 2)], avg @[(7, 1)] | no parameter-free ops. | 87.70% |
| 2 | 3 | max @[(2, 1), (5, 1;2)] | no parameter-free ops. | 87.65% |
| 3 | 1 | max @[(5, 1;2;3), (7, 1;2;3)] | max @[(3, 3), (6, 3)] | 88.09% |
| 3 | 2 | max @[(4, 2), (5, 2), (7, 2)], avg @[(7, 1)] | max @[(3, 3), (6, 3)] | 88.06% |
| 3 | 3 | max @[(4, 1;3), (5, 3)], avg @[(5, 2)] | no parameter-free ops. | 88.03% |
| 4 | 1 | max @[(2, 3;4), (5, 1;2;3;4), (7, 1)], avg @[(8, 1;2)] | avg @[(3, 1;2), (6, 1;2)] | 88.12% |
| 4 | 2 | max @[(4, 1;3;4), (5, 2;3;4), (7, 1;2;3;4)], avg @[(5, 1)] | no parameter-free ops. | 87.79% |
| 4 | 3 | max @[(2, 3), (7, 1;2)], avg @[(4, 1;2;3), (8, 2)] | avg @[(3, 2), (6, 2)] | 87.54% |

**On Neural Architecture Searches.**  We further investigate whether parameter-free operations are likely to be chosen for model accuracy along with trainable operations in a neural architecture search (NAS). We choose DARTS (Liu et al., 2019) with the default architecture of eight cells and perform searches on CIFAR-10 (Krizhevsky, 2009)[2]. Towards a more sophisticated study, we refine the search configuration as follows. First, we force the entire normal cells to be searched individually rather than following the default setting searching one unified normal cell. This is because when searching with the unified normal cell, it is more likely to be searched without the parameter-free operations to secure the expressiveness of the entire network (see the search result

---

[2]We follow the original training settings of DARTS (Liu et al., 2019). Additionally, to avoid the collapse of DARTS due to the domination of skip connections in learned architectures, we insert the dropout at the SKIP_CONNECT operation proposed in the method (Chen et al., 2019b).

at the first row in Table 1). Therefore we prevent this with the more natural configuration. Second, the operation list is changed to ensure fairness of the expressiveness among the operations; since the primitive DARTS has the default operation list with the separable convolutions (SEP_CONV_3×3, SEP_CONV_5×5) and the dilated separable convolutions (DIL_CONV_3×3, DIL_CONV_5×5) which consist of multiple convolutions and ReLU with BN. Thus, we need to simplify the primitive operations to [MAX_POOL_3×3, AVG_POOL_3×3, CONV_1×1, CONV_3×3, DW_CONV_3×3, ZERO, SKIP_CONNECT]. Finally, we further perform searches with different numbers of nodes inside each cell to confirm the trend of search results regarding the architectural constraint. All the searches are repeated with three different seed numbers.

As shown in Table 1 and Fig.3, we observe the parameter-free operations are frequently searched in the normal cells, which is different from the original search result (at the first row) similarly shown in the previous works (Liu et al., 2019; Chen et al., 2019b; Liang et al., 2019; Chu et al., 2020). Moreover, the accuracies are similar to the original setting with two unified cells, where the parameter-free operations are not searched in the normal cell (see the first and the last rows). Interestingly, the number of the picked parameter-free operations increases as the number of nodes increases; they do not dominate the reduction cells. As a result, it turns out that the accuracy objective in searches lets the parameter-free operations become chosen, and therefore the operations can be used at certain positions which looks similar to the Inception

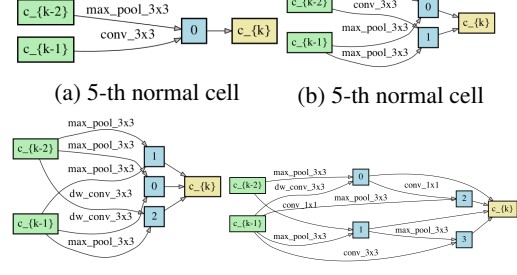

(a) 5-th normal cell     (b) 5-th normal cell

(c) 7-th normal cell     (d) 5-th normal cell

Figure 3. **Visualization of the searched cells.** The example searched cells shown in Table 1 are visualized.

blocks (Szegedy et al., 2015; Ioffe & Szegedy, 2015; Szegedy et al., 2016). Notice that similar results are produced with different settings and seeds.

# 4 DESIGNING EFFICIENT DEEP NEURAL NETWORKS

Based on the studies above, parameter-free operations can be a building block of designing a network architecture. We apply the parameter-free operations to redesign a deeper neural network architecture, including a convoluption neural network (CNN) and a vision transformer (ViT).

## 4.1 EFFICIENT CNN ARCHITECTURES

**Hybrid Architecture with Efficient Bottlenecks.** We employ the proposed efficient bottleneck to design deeper efficient CNNs. The max-pool operation plays a role of the spatial operation in the efficient bottleneck; specifically, the 3×3 convolution, BN, and ReLU triplet are replaced with a max-pool operation[3]. To design the entire network architecture, fully utilizing the efficient bottlenecks would yield a much faster inference speed model, but a combination of the efficient bottleneck and the regular bottleneck has a better trade-off between accuracy and speed. We use ResNet50 (He et al., 2016b) as the baseline to equip a simple heuristic of building a hybrid architecture of the efficient bottleneck and the regular bottleneck. The observation from the NAS searches in §3.3, the parameter-free operations are not explored much at the downsampling blocks but normal blocks. Thus, we similarly design a new efficient network architecture using the efficient bottlenecks as the fundamental building block except for the downsampling positions with the regular bottlenecks. We call this model *hybrid* architecture. The details of the architectures are illustrated in Appendix A.

**Architectural Study.** The architectural study is further conducted with the replacements of the regular spatial convolutions in the regular bottlenecks. We involve the efficient bottlenecks into the stages of ResNet50, replacing the existing regular bottlenecks exhaustively to cover the design space as much as possible. It is worth noting that performing a neural architecture search would yield a precise design guide, but a direct search on ImageNet is costly. We use the ImageNet dataset to provide a more convincing result of the investigation. We also report the performance of the competitive

---

[3]We found that parameter-free operations such as the max-pool operation without following BN and ReLU do not degrade accuracy. Mathematically, using ReLU after the max-pool operation (without BN) does not give nonlinearity due to the preceding ReLU. Empirically, involving BN after a parameter-free operation could stabilize the training initially, but model accuracy was not improved in the end.

architectures replacing the spatial operations (*i.e.*, the 3×3 convolutions) with 1) the 1×1 convolutions; 2) the 3×3 depthwise convolution in entire stages.

Table 2 shows the trade-off between accuracy and speed of the different models mixing the regular and the efficient bottlenecks. **B→B→B→B** denotes the baseline ResNet50; **E / B** denotes the model using the regular bottleneck and the efficient block alternately. Among the different models, we highlight *hybrid* and **E / B** models due to the promising performance. This shows that only a simple module-based design regime can achieve improved performance over the baseline. **E→E→E→E** is the model using the efficient bottlenecks only and shows the fastest speed as we expected. The model with the 1×1 convolution replacements cannot reach the accuracy standard presumably due to the lack of the receptive field; this is why a sufficient number of spatial operations are required. We observe that the model with the depthwise convolutions shows a promising performance, but our models surpass it as we similarly observed in §3.3. All the models are trained on ImageNet with the standard 90-epochs training setup[4] (He et al., 2016a) to report the performance.

Table 2. **Model study on ResNet50**. We report the performance of *hybrid* model, which is the most promising model compared with others, and the models with stage-level combinations of the regular bottleneck and the efficient bottleneck (denoted by **B** and **E**, respectively).We further report the performance of two variant models by replacing each 3×3 convolution with 1) a 1×1 convolution; 2) a 3×3 depthwise convolution in all the regular bottlenecks. The two best trade-off models are in **boldface**.

| Models | GPU Lat. (ms) | Top-1 acc.(%) |
|---|---|---|
| *Hybrid* | **7.3** | **74.9** |
| **E / B** | **7.7** | **75.3** |
| **B→B→B→B** | 8.7 | 76.2 |
| **E→B→B→B** | 8.4 | 75.4 |
| **E→E→B→B** | 8.0 | 75.1 |
| **E→E→E→B** | 7.3 | 73.1 |
| **E→E→E→E** | 6.8 | 72.0 |
| **B→B→B→E** | 8.4 | 75.4 |
| **B→B→E→E** | 7.7 | 74.6 |
| **B→E→E→E** | 7.3 | 74.0 |
| 1×1 conv | 6.2 | 30.1 |
| 3×3 dwconv | 8.3 | 74.9 |

### 4.2 EFFICIENT ViT ARCHITECTURES

We provide a further use case of the parameter-free operations in a totally different architecture from CNN. We again choose the max-pool operation as the parameter-free operation and apply it to replace the self-attention layer (Vaswani et al., 2017) in vision transformer (ViT) (Dosovitskiy et al., 2021) to observe how the efficient operation can replace a complicated layer. We do not change the MLP in a transformer block but replace the self-attention module with the max-pool operation. Specifically, after the linear projection with the ratio of 3 (*i.e.*, identical to the concatenation of the projection of query, key, and value in the self-attention layer in ViT), the projected 1d-features are reshaped to 2d-features for the input of the spatial operation. Consequently, the self-attention layer is replaced with a cheaper parameter-free operation and will bring clear efficiency. We use the global average pooling (GAP) layer instead of using the classification token of ViT since the classification token is hardly available without the self-attention layer. Many transformer-based ViTs can be a baseline; we additionally adopt a strong baseline Pooling-based Vision Transformer (PiT) (Heo et al., 2021b) to show applicability. More details are elaborated in Appendix A.

## 5 EXPERIMENTS

### 5.1 IMAGENET CLASSIFICATION

**Efficient ResNets.** We perform ImageNet (Russakovsky et al., 2015) trainings to validate the model performance. We adopt the standard architecture ResNet50 (He et al., 2016b) as the baseline and train our models with the aforementioned standard 90-epochs training setting to fairly compare with the competitors (Luo et al., 2017; Huang & Wang, 2018; Wu et al., 2018; Wang et al., 2018; Han et al., 2020; Yu et al., 2019; Qiu et al., 2021), where the efficient operators were proposed or the networks were pruned. We report each averaged speed of the models with the publicly released codes on a V100 GPU. Table 3 shows our networks acheieve faster inference speeds than those of the competitors with the comparable accuracies. The channel-width pruned models (Slimmable-R50 0.5×, 0.75×) and the models with new operations (Veratile-R50, GhostNet-R50, and SlimConv-R50) can-

---

[4]Trainings are done with the fixed image size 224×224 and the standard data augmentation (Szegedy et al., 2015) with the random_resized_crop rate from 0.08 to 1.0. We use stochastic gradient descent (SGD) with Nesterov momentum (Nesterov, 1983) with momentum of 0.9 and mini-batch size of 256, and learning rate is initially set to 0.4 by the linear scaling rule (Goyal et al., 2017) with step-decay learning rate scheduling; weight decay is set to 1e-4. The accuracy of the baseline ResNet50 has proven the correctness of the setting.

Table 3. **ImageNet performance comparison of efficient models.** We report the model performance including accuracy, the number of parameters, FLOPs, and the GPU latency measured on a V100 GPU. All the model speeds are measured by ourselves using the publicly released architectures. [†]: used further training recipes.

| Network Architecture | Params. (M) | FLOPs (G) | GPU (ms) | Top-1 (%) | Top-5 (%) |
|---|---|---|---|---|---|
| ResNet50 (R50) (He et al., 2016b) | 25.6 | 4.1 | 8.7 | 76.2 | 93.8 |
| Thinet70-R50 (Luo et al., 2017) | 16.9 | 2.6 | - | 72.1 | 90.3 |
| SSS-R50 (Huang & Wang, 2018) | 18.6 | 2.8 | - | 74.2 | 91.9 |
| Shift-R50 (Wu et al., 2018) | 22.x | N/A | - | 75.6 | 92.8 |
| Versatile-R50 (Wang et al., 2018) | 17.1 | 1.8 | 18.7 | 75.5 | 92.4 |
| Sllimable-R50 ($0.5\times$) (Yu et al., 2019) | 6.9 | 1.1 | 8.5 | 72.1 | N/A |
| Sllimable-R50 ($0.75\times$) (Yu et al., 2019) | 14.8 | 2.4 | 8.6 | 74.9 | N/A |
| GhostNet-R50 (Han et al., 2020) | 14.0 | 2.1 | 20.3 | 75.0 | 92.3 |
| SlimConv-R50 ($k=8/3$) (Qiu et al., 2021) | 12.1 | 1.9 | 24.5 | 75.5 | N/A |
| Ours-R50 (max) | 14.2 | 2.2 | 6.8 | 72.0 | 90.5 |
| Ours-R50 (*hybrid*) | 17.3 | 2.6 | 7.3 | 74.9 | 92.2 |
| Ours-R50 (deform_max) | 18.0 | 2.9 | 10.3 | 75.3 | 92.5 |
| Ours-R50 (max)[†] | 14.2 | 2.2 | 6.8 | 74.3 | 92.0 |
| Ours-R50 (*hybrid*)[†] | 17.3 | 2.6 | 7.3 | 77.1 | 93.1 |
| Ours-R50 (deform_max)[†] | 18.0 | 2.9 | 10.3 | 78.3 | 93.9 |

Table 4. **ImageNet performance of CNN models.** We report the ImageNet performance, mCE (ImageNet-C) and AUC (ImageNet-O) of the diverse CNN models. All the redesigned models experience massive reductions of the computational costs and the substantial gains on mCE and AUC but barely drop the accuracy ($<2.0\%$).

| Network | Params. (M)↓ | FLOPs (G)↓ | GPU (ms)↓ | CPU (ms)↓ | Top-1 (%)↑ | Top-5 (%)↑ | mCE (%)↓ | AUC (%)↑ |
|---|---|---|---|---|---|---|---|---|
| ResNet50 | 25.6 | 4.1 | 8.7 | 45.4 | 78.5 | 94.2 | 63.8 | 51.7 |
| Ours-R50 | 17.3 (-33%) | 2.6 (-37%) | 7.3 (-17%) | 39.8 (-12%) | 77.1 (-1.4) | 93.1 (-1.1) | 57.5 (-6.3) | 52.9 (+1.2) |
| ResNet50-SE | 28.1 | 4.1 | 13.9 | 98.0 | 79.5 | 94.7 | 69.6 | 58.9 |
| Ours-R50-SE | 19.9 (-29%) | 2.6 (-37%) | 12.5 (-10%) | 92.6 (-9%) | 78.2 (-1.3) | 93.9 (-0.8) | 63.8 (-5.8) | 59.2 (+0.3) |
| ResNet101 | 44.6 | 7.8 | 16.7 | 80.5 | 80.1 | 94.9 | 69.7 | 51.7 |
| Ours-R101 | 26.3 (-41%) | 4.3 (-45%) | 13.5 (-19%) | 65.7 (-18%) | 78.2 (-1.9) | 93.8 (-1.1) | 62.0 (-7.7) | 54.6 (+2.9) |
| WRN50-2 | 68.9 | 11.4 | 9.0 | 84.0 | 79.7 | 94.7 | 67.0 | 49.8 |
| Ours-WRN50-2 | 36.0 (-48%) | 5.4 (-53%) | 7.3 (-20%) | 59.8 (-29%) | 78.1 (-1.6) | 93.8 (-0.9) | 60.9 (-6.1) | 51.9 (+2.1) |
| WRN101-2 | 126.9 | 22.8 | 16.9 | 146.2 | 80.9 | 95.3 | 73.2 | 50.6 |
| Ours-WRN101-2 | 53.9 (-58%) | 8.9 (-61%) | 13.7 (19%) | 96.3 (-34%) | 78.9 (-2.0) | 94.2 (-1.1) | 63.4 (-9.8) | 54.8 (+4.2) |

not reach the model speed to ours. Additionally, we report the improved model accuracy with further training tricks (see the details in Appendix C). This aims to show the maximal capacity of our model even using many parameter-free operations inside. The last rows in the table present our model can follow the baseline accuracy well, and we see the gap between the baseline and ours has diminished; this show a potential of using parameter-free operations for a network design.

**Bigger Network Architectures.** Our design regime is applied to complicated network architectures such as ResNet50-SE (Hu et al., 2017) and Wide ResNet-101-2 (WRN101-2) (Zagoruyko & Komodakis, 2016). We report the ImageNet performance plus mean Corruption Error (mCE) on ImageNet-C (Hendrycks & Dietterich, 2019) and Area Under the precision-recall Curve (AUC) the measure of out-of-distribution detection performance on ImageNet-O (Hendrycks et al., 2021). Table 4 indicates the models redesigned with the parameter-free operations work well; bigger models are more significantly compressed in the computational costs. Additionally, we notice all the mCEs and AUC of our models are remarkably improved, overtaking the accuracy degradation, which means the models with parameter-free operations suffer less from overfitting.

**Deformable Max-pool Operation.** We manifest a future direction of utilizing a parameter-free operation. We borrow a similar idea of the deformable convolution[5] (Dai et al., 2017; Zhou & Feng, 2017). Specifically, we identically involve a convolution to interpolate the features predicted by itself to perform a parameter-free operation on. The max-pool operation still covers the spatial operation; only the offset predictor has weights to predict the locations. We regard this operator as to how the performance of the max-pool operation can be stretched when involving few numbers of parameters. Only performing computation on predicted locations can improve the accuracy over the vanilla

---

[5]We implement the operation upon the code: https://github.com/CharlesShang/DCNv2.

Table 5. **COCO object detection results.** All the models are finetuned on `train2017` by ourselves using the ImageNet-pretrained backbones in Table 2. We report box APs on `val2017`.

| Backbone | IN Acc. (%) | Input Size | Bbox AP at IOU | | | GPU (ms) | Params. (M) | FLOPs (G) |
|----------|-------------|------------|------|-----------|-----------|----------|-------------|-----------|
| | | | AP | AP$_{50}$ | AP$_{75}$ | | | |
| ResNet50 | 76.2 | 1200×800 | 32.9 | 51.8 | 35.1 | 42.8 | 41.8 | 202.2 |
| ResNet50 (*hybrid*) | 74.9 | 1200×800 | 31.9 (-1.0) | 51.5 | 34.1 | 37.5 (-12%) | 33.6 (-20%) | 175.8 (-13%) |
| ResNet50 (deform_max) | 75.3 | 1200×800 | 33.2 (+0.3) | 53.0 | 35.3 | 47.7 (+11%) | 34.3 (-18%) | 181.8 (-10%) |

Table 6. **ImageNet performance of ViTs.** We report ViT models' performance trained on ImageNet with diverse training settings denoted by Vanilla, CutMix, and DeiT (with strong augmentations).

| Model | Throughput (imgs/sec) | | Vanilla | +CutMix | +DeiT |
|-------|-----------|-----------|---------|---------|-------|
| | 256-batch | 1-batch | | | |
| ResNet50 | 962 | **112** | 76.2 | 77.6 | 78.8 |
| ViT-S | 787 | 86 | 73.9 | 77.0 | 80.6 |
| ViT-S (dw-conv) | 571 (-216) | 95 (+9) | 76.1 (+2.2) | 78.7 (+1.7) | 81.2 (+0.6) |
| ViT-S (max-pool) | 763 (-24) | 96 (+10) | 74.2 (+0.3) | 77.3 (+0.3) | 80.0 (-0.6) |
| PiT-S | 952 | 57 | 75.5 | 78.7 | 81.1 |
| PiT-S (dw-conv) | 781 (-171) | 90 (+33) | 76.1 (+0.5) | 78.6 (-0.1) | 81.0 (-0.1) |
| PiT-S (max-pool) | **1000** (+48) | 92 (+35) | 75.7 (+0.2) | 78.1 (-0.6) | 80.8 (-0.3) |

computation with a few extra costs, as shown in Table 3. Note that there is room for faster operation speed as the implementation can be further optimized.

## 5.2 COCO OBJECT DETECTION

We verify the transferability of our efficient backbones on the COCO2017 dataset (Lin et al., 2014). We adopt the standard Faster RCNN (Ren et al., 2015) with FPN (Lin et al., 2017) without any bells and whistles to finetune the backbones following the original training settings (Ren et al., 2015; Lin et al., 2017) Table 5 shows our models achieve a better trade-off between the AP scores and the computational costs, including the model speed, and does not significantly degrade the AP scores even inside massive parameter-free operations. We further validate the backbone with the deformable max-pool operations in Table 5. Strikingly, AP scores are improved over ResNet50 even with the low ImageNet accuracy; this shows the effectiveness of the operation in a localization task.

## 5.3 IMAGENET CLASSIFICATION WITH EFFICIENT VISION TRANSFORMERS

We demonstrate using parameter-free operations in ViT (Dosovitskiy et al., 2021) in a novel way. We follow the aforementioned architectural modifications. Two vision transformer models ViT-S and PiT-S (Heo et al., 2021b) are trained on ImageNet with three different training settings: Vanilla (Dosovitskiy et al., 2021), with CutMix (Yun et al., 2019) and DeiT (Touvron et al., 2021) settings in Heo et al. (2021b). We similarly apply the depthwise convolution into the self-attention layer, which can be a strong competitor to make a comparison. We report the performance of the models in Table 6; we report throughput (images/sec) as the speed measure following the ViT papers (Dosovitskiy et al., 2021; Heo et al., 2021b), which is measured on 256 batch-size and a single batch-size both. The result demonstrates that ViT and PiT with the max-pool operation have faster throughput without significant accuracy degradation compared with the baselines; the depthwise convolution is a promising alternative, but the throughput is a matter compared with the max-pool operation for both architectures. Interestingly, PiT takes advantage of using the parameter-free operation, which presumably comes from larger features in early layers.

## 6 CONCLUSION

In this paper, we rethink parameter-free operations as the building block of learning spatial information to explore a novel way of designing network architecture. We have experimentally studied the applicability of the parameter-free operations in network design and rebuild network architectures, including convolutional neural networks and vision transformers, towards more efficient ones. Extensive results on a large-scale dataset including ImageNet and COCO with diverse network architectures have demonstrated the effectiveness over the existing efficient architectures and the use case of the parameter-free operations as the main building block. We believe our work highlighted a new design paradigm for future research beyond conventional efficient architecture designs.

ACKNOWLEDGEMENTS

We would like to thank NAVER AI Lab members for valuable discussions. We also thank Seong Joon Oh, Sangdoo Yun, and Sungeun Hong for peer-reviews. NAVER Smart Machine Learning (NSML) (Kim et al., 2018) has been used for experiments.

ETHICS STATEMENT

This paper studies a general topic in computer vision which is designing an efficient network architecture. Therefore, our work does not be expected to have any potential negative social impact but would contribute to the computer vision field by providing pretrained models.

REPRODUCIBILITY STATEMENT

We provide detailed information of all the experiments in the paper. Furthermore, the details of our models with specific training hyper-paramaters are clearly announced for those who would like to design or train our proposed models.

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

APPENDIX

## A    DETAILS OF EFFICIENT ARCHITECTURES

We elaborate on the efficient building blocks used in the experiments above. Fig.A.1 shows the schematic illustration of the proposed blocks compared with the original ones. We observe that the modification is simple and readily be applied to any network architecture.

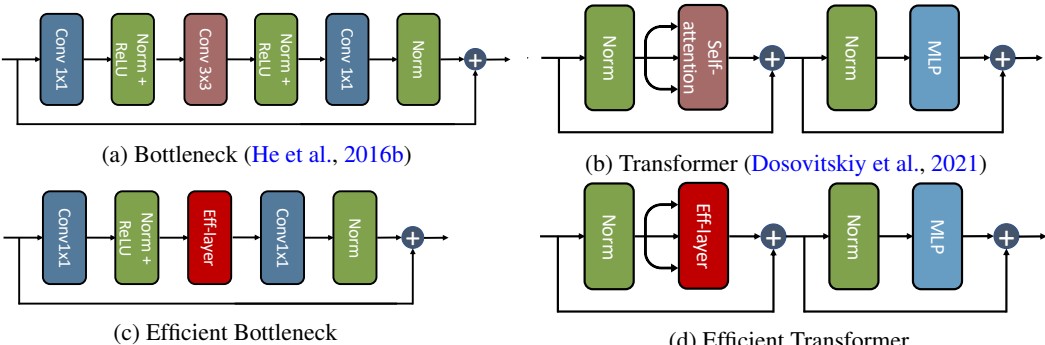

(a) Bottleneck (He et al., 2016b)

(b) Transformer (Dosovitskiy et al., 2021)

(c) Efficient Bottleneck

(d) Efficient Transformer

Figure A.1. **Schematic illustration of the efficient building blocks.** We visualize (a) the regular bottleneck in ResNets (He et al., 2016b); (b) the regular transformer in ViTs (Dosovitskiy et al., 2021); (c) our efficient bottleneck; (d) our efficient transformer; the eff-layers in (c) and (d) denote the parameter-free operations.

**Efficient Bottleneck.**    We replace the triplet of $3\times3$ convolution, BN (Ioffe & Szegedy, 2015), and ReLU (Nair & Hinton, 2010) in the regular bottleneck with a single spatial parameter-free operation (denoted as eff-layer in Fig.A.1c). The fastest architecture in Table 2 and Table 3 fully replace the regular bottlenecks, including the downsampling blocks with the efficient bottlenecks; therefore, the parameter-free operation (here the max-pool operation) plays a role of spatially aggregating the features by reducing the resolution. The *hybrid* architecture has an almost identical design regime with the most efficient architecture (*i.e.*, $\mathbf{E}\rightarrow\mathbf{E}\rightarrow\mathbf{E}\rightarrow\mathbf{E}$ in Table 2) except for the downsampling blocks. We remain each downsampling block with the regular bottleneck blocks and involve an efficient parameter-free operation before the spatial operation (we assign the avg-pool as the parameter-free operation). We apply the identical bottleneck configuration of the *hybrid* architecture to the diverse deep convolutional neural networks (CNNs) in Table 4. Note that we do not modify the depth and the width of network architecture, the stem of CNNs that is the first set of layers before the first bottleneck, and the output layer having a fully connected layer with 1000-d output dimension. Furthermore, we do not modify specific architectural elements such as the SE-block of ResNet50-SE (Hu et al., 2017) inside each regular bottleneck but use it in each efficient bottleneck at the same positions.

**Efficient Transformer.**    Designing the efficient vision transformer (ViT) architecture is done by replacing half of the self-attention layers with the spatial parameter-free operation (denoted as eff-layer in Fig.A.1d). In other words, the original transformer block and the efficient transformer block are used alternately. We use the $3\times3$ max-pool operation for eff-layer in the efficient transformer block and compare with the variant using the $5\times5$ depth-wise convolution for eff-layer. ReLU is added after eff-layer and the first linear layer of the efficient transformer block to give additional non-linearity. For Pooling-based Vision Transformer (PiT) (Heo et al., 2021a) and ViT (Dosovitskiy et al., 2021), we do not modify the architectural elements including 1) the patch size; 2) the stem that patchifies the input for the following transformer; 3) the number of transformers. We only change the classification head position from the classification token to the Global Average Pooling (GAP) at the last layer since the classification token is not compatible with convolutional operations. As reported in Zhai et al. (2021), transformer with GAP shows comparable performance to ViT with the classification token. We also evaluate a more efficient network architecture which is fully equipped with the efficient transformer; it achieves faster speed, but slightly less accurate. We leave further improvements with a parameter-free operation inside the efficient transformer to reach the accuracy of the original transformer as future work.

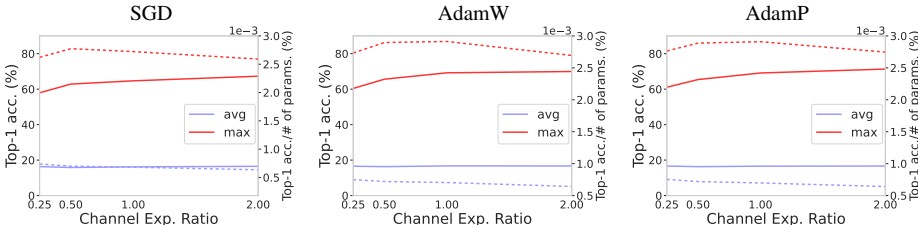

Figure B.1. **Comparison of the parameter-free operations.** We visualize top-1 accuracy (**solid lines**) with accuracy per # parameters (**dashed lines**) similar to Fig.1 with two parameter-free operations. All the settings are identical to the previous study, so we only plot the channel width of 32. We observe the max-pool operation consistently beat the avg-pool operation in a single bottleneck training.

## B    ON PARAMETER-FREE OPERATIONS

**Can We Use Avg-pool as the Spatial Parameter-free Operation?**    We mainly used the max-pool operation as the spatial parameter-free operation that replaces the regular convolutions and the self-attention layer in experiments. The max-pool operation is expected to have higher expressiveness compared with that of the avg-pool operation. Because the avg-pool operation is a conceptually smoothing operation, and the max-pool operation contains a nonlinearity similar to what ReLU has (*i.e.*, $\max(x)$ and $\max(x,0)$). Experimentally, we found low expressiveness of the avg-pool operation in the shallow network study. Fig.B.1 shows the comparison of the max-pool and the avg-pool operations trained in a single bottleneck block. The setting is identical to the previous study in §3, and we only visualize the case of the channel width of 32 because a similar trend with the channel width of 64 was observed. We observe the large accuracy gaps between the two operations. This may be a ground of the search results where few avg-pool operations are chosen in normal cells in the previous NAS experiments in §3.3.

**Deformable Avg-pool Operation.**    We here provide an interesting idea of using the avg-pool operation, which shows inferior outcomes than the max-pool operation in the operation expressiveness. We have proposed the deformable max-pool operation involving a few parameters to improve the parameter-free operation's discriminative power significantly. Using the same idea into the avg-pool operation is available; surprisingly, a downside of the low expressiveness of the avg-pool operation has been vanished, as shown in Table B.1. This result exhibits that even a smoothing operation can be employed to learn discriminative features in a deep neural network.

Table B.1. **ImageNet performance of deformable operations.** We report the model performance trained on ImageNet for the deformable operations. The new operation, dubbed deformable avg-pool, surprisingly reaches the accuracy of the deformable max-pool operation.

| Network Architecture | Params. (M) | FLOPs (G) | GPU (ms) | Top-1 (%) | Top-5 (%) |
|---|---|---|---|---|---|
| Ours-R50 (deform_max) | 18.0 | 2.9 | 10.3 | 78.3 | 94.0 |
| Ours-R50 (deform_avg) | 18.0 | 2.9 | 10.3 | 78.3 | 93.9 |

## C    DETAILED TRAINING RECIPES

We use the following training recipes to maximize the accuracy of the ResNet-based models. We use the cosine learning rate scheduling (Loshchilov & Hutter, 2017a) with the initial learning rate of 0.5 using four V100 GPUs with batch size of 512. Exponential moving average (Tarvainen & Valpola, 2017) over the network weights is used during training. We use the regularization techniques and data augmentations including label smoothing (Szegedy et al., 2016) (0.1), RandAug (Cubuk et al., 2019) (magnitude of 9), Random Erasing (Hermans et al., 2017) with pixels (0.2), lowered weight decay (1e-5), and a large training epochs (400 epochs)[6]. We use the code baseline in the renowned repository[7] for our ImageNet training. The accuracy of the baseline models could reach the known

---

[6]When training for a larger epochs (600 epochs), the top-1 accuracy of Ours-R50 (max) in Table 3 is improved to 75.5%, which gets closer to the original ResNet's; furthermore, Ours-R50 (*hybrid*), and Ours-R50 (deform_max) reach 78.0 and 79.3, respectively.

[7]https://github.com/rwightman/pytorch-image-models/

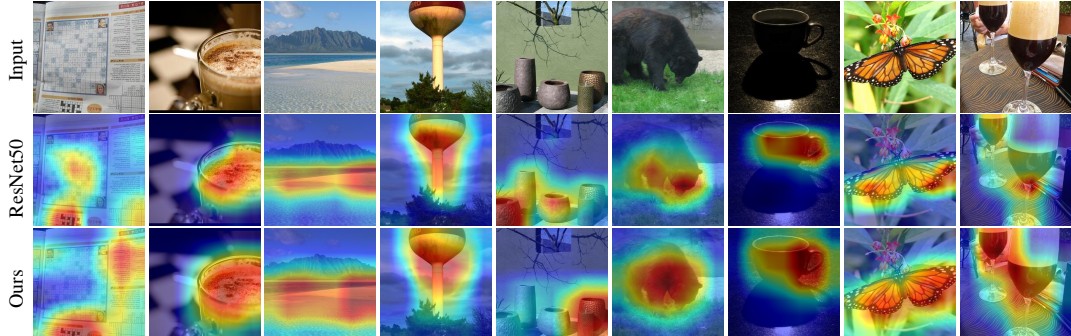

Figure D.1. **Grad-CAM visualization of the final features.** We visualize the highlighted features from the output of the final stage using Gradient-weighted Class Activation Mapping (Grad-CAM) (Selvaraju et al., 2017). The first row shows the original images, which are randomly picked from the ImageNet validation set; the second row shows the visualized features of the ImageNet-pretrained ResNet50; the last row shows the result of our ImageNet-pretrained ResNet50 (*hybrid*). Ours show similar outputs compared with ResNet50 but seem to capture the foreground with wider regions.

improved accuracy reported in such a paper (Bello et al., 2021) which presents the training recipes for highly improved models.

## D UNDERSTANDING MAX-POOL OPERATION

This section provides intuitive explanations how parameter-free operations such as the max-pool operation could work as replacing a trainable layers in a network.

### D.1 CONNECTION WITH MAXOUT

Maxout (Goodfellow et al., 2013) performs the max operation to a set of the inputs like an activation function with the multiple inputs. Maxout is designed to use after a bunch of linear or convolution layers to ensemble the outputs in a nonlinear manner, which leads to the increase of model capacity. The accuracy improvement in the original paper (Goodfellow et al., 2013) results from the claim that Maxout can approximate any operations, namely perform as a universal approximator. From an architectural point of view, the success of Maxout is probably due to the enhanced model capacity by the increased input dimension with the nonlinearity imposed by the max operation.

For the max-pool operation with the efficient bottleneck mainly used throughout the paper, the relationship between the Maxout operation and ours is worth discussing. Maxout outputs through the max operation of the outputs of multiple linear layers computed from a single input. Otherwise, the max-pool operation in the efficient bottleneck performs the max operation with multiple transformed inputs (i.e., transformed pixels spatially adjacent to each point) by the preceding $1\times1$ convolution, which acts as a single linear layer for a given channel. However, if we replace the small variations in neighboring pixels with the perturbations in the weights of the linear layer as claimed in Seong et al. (2018), then we may interpret the max-pool operation performs like Maxout.

### D.2 RECEPTIVE FIELD MATTER

Since the spatial parameter-free operations have the receptive field (Luo et al., 2016) like convolutions, we conjecture such simple operations can capture striking pixels (or patches). We empirically support the conjecture by providing a visualization of the final output features of the ImageNet-pretrained ResNet50 and our ResNet50 (*hybrid*) in Fig.D.1. We observe that two models successfully localize the foreground object in common for all the images. A noticeable difference is that ResNet50 tends to focus more on specific crucial regions but ResNet50 (*hybrid*) on wider regions (a similar trend will be observed in Fig.D.2). Note that the area of the highlighted region does not directly link to a model's accuracy (in fact, ResNet50 has higher accuracy). We believe that this indicates how much a model learns localizable features and can be utilized to understand the learning dynamics of various models.

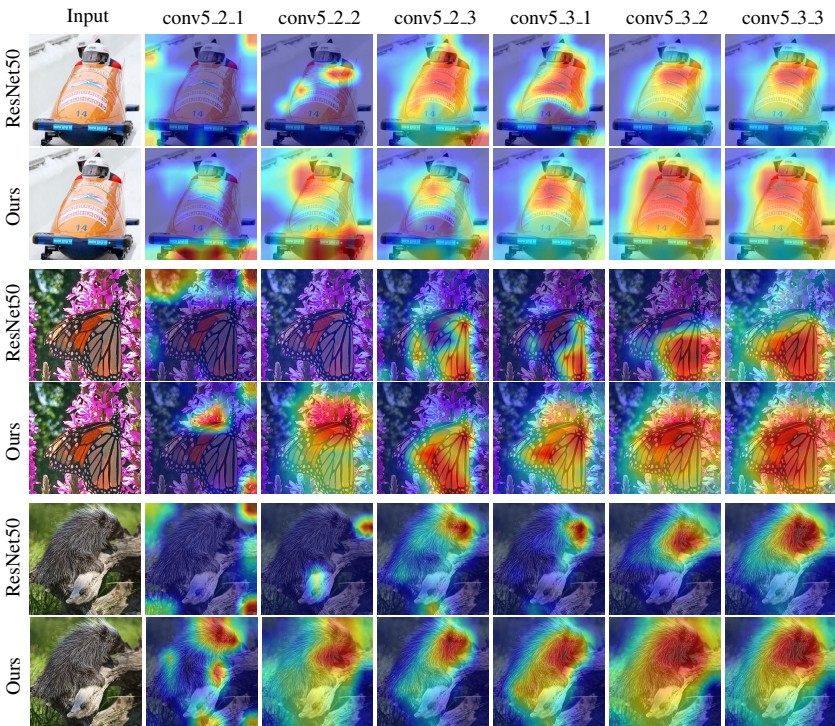

Figure D.2. **Grad-CAM visualization of intermediate features.** We visualize the highlighted features by Grad-CAM of the intermediate features produced by the six successive layers in the last two bottlenecks of the final stage in ResNet50 and ResNet50 (*hybrid*), respectively. Each visualized feature from left to right gets closer to the final output features of a model. Each feature of conv5_x_y denotes the output features of the y-th layer in the x-th bottleneck, where a bottleneck has the 1×1 convolutions at the 1st and 3rd layers) and 3x3 convolution/the max-pool operation at the 2nd layer in ResNet50/our ResNet50 (*hybrid*). The input images are randomly picked from the ImageNet validation set.

We further conjecture that a 1×1 convolution can complement the expressiveness of a parameter-free operation such as the max-pool operation or avg-pool operation by nonlinearly mixing the features, which is computed by the parameter-free operation. To make a ground for the conjecture, we would like to visualize the intermediate features extracted from 1) each of two 1×1 convolutions; 2) the spatial operations including the 3×3 convolution and the max-pool operation in a bottleneck block. We choose the last two bottlenecks in the final stage conv5 and visualize the output features of the aforementioned layers.

Fig.D.2 shows the highlighted features produced by the three different input images randomly picked from the ImageNet validation set. We use the identical models employed to visualize the final features shown in Fig.D.1. We let conv5_x_y denote the output features of the y-th layer in the x-th bottleneck of a specific model; for example, each visualized feature of conv5_2_2 in ResNet50 and our model (ResNet50 (*hybrid*)) indicates the output features of the 3×3 convolution and the spatial max-pool operation, respectively. First, we observe the 1×1 convolutions refine the features to be more discriminative; when comparing the features in conv5_x_2 with those of conv5_x_3, the output features usually get more highlighted on the crucial region in the foreground objects. Furthermore, it seems that the 1×1 convolutions change more of the previous features in our model, which may come from the different output features by the 3×3 convolutions and the max-pool operations (compare the output features of ResNet50 with ours in conv5_x_2). Therefore, based on the observations, the 1×1 convolution performs to make features more discriminative, and they do more with the parameter-free operations like the max-pool operation.

# E  MORE VISUALIZATIONS

In this section, we further discuss the studied materials with the additional graphs and figures to provide more detailed information. §E.1 contains extended trade-off graphs of our study in §3.3;

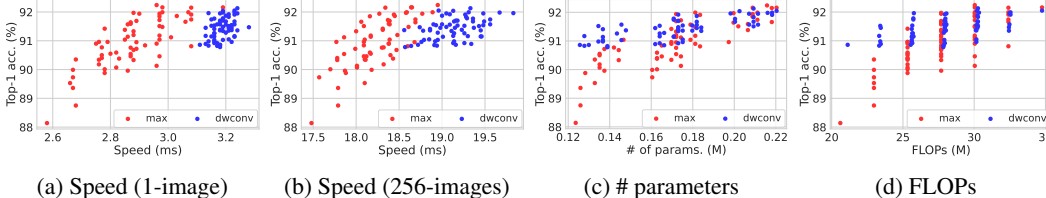

| (a) Speed (1-image) | (b) Speed (256-images) | (c) # parameters | (d) FLOPs |

Figure E.1. **Multiple bottlenecks study (cont'd).** The entire models trained in the multiple bottlenecks study in §3.3 are visualized in the comparison graphs: (a) Accuracy vs. speed with the batch size of 1; (b) Accuracy vs. speed with the batch size of 256; (c) Accuracy vs. # parameters; (d) Accuracy vs. FLOPs. The graphs includes the 20% best-performing models plotted in Fig.2. The depthwise convolution (blue dots) shows efficiency again in # parameters and FLOPs, but the max-pool (i.e., red dots) has clear benefits in the speed measures. Note that the speed gap between the two operations gets larger when processing multiple images, as shown through (a) and (b).

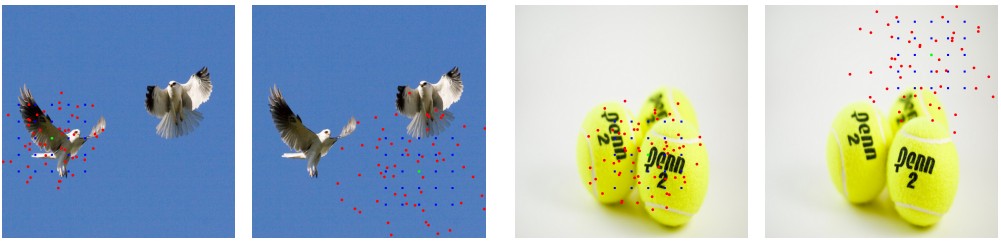

Figure E.2. **Visualization of predicted offsets.** We illustrate the predicted offsets by our deformable max-pool operation from the images from the validation set in ImageNet. Each sampled image has two predicted offsets on a foreground object (left) and the background (right). We plot 1) the predicted offsets (red dots); 2) the initial grid-like points (blue dots) with the center (green dots) in each figure.

§E.2 discusses predicted offsets by the proposed deformable max-pool operation; finally in §E.3, we discuss the performance trade-offs of the models on the ImageNet datasets in §5.1.

### E.1    Multiple Bottlenecks Study (cont'd)

Fig.E.1 shows the performance of the entire models which are trained in §3.3. The graphs includes the 20% best-performing models shown in Fig.2. As expected, the models using many max-pool operations have degraded accuracies but show clear speed benefits; there is about a 1.5% accuracy gap between the fastest models using each operation yet more than a 5ms speed gap between them.

### E.2    Predicted Offsets by Deformable Max-pool Operation

We visualize the predicted offsets by the proposed deformable max-pool operation (deform_max) with the sample images in Fig.E.2. We plot the aggregated offsets of the last two deform_maxs in the final stage of ResNet50 with the initial points, which look like the grid-like offset points of the two successive regular convolutions. We observe the offsets are concentrated on the objects when the center is on each foreground object. The offsets spread widely when the center is on the background, which is similarly observed in the deformable convolution paper (Dai et al., 2017; Zhu et al., 2019). It is surprising that our models only trained with the ImageNet's class labels without strong supervisions (i.e., detection boxes or segmentation masks) show the improved localization capability. Furthermore, albeit our model does not be trained with the background class, each shape of the predicted offsets on foreground and background looks quite different as shown in Fig.E.2.

We would like to stress the main differences of the experimental settings with those in the deformable convolution papers. First, our model is trained on ImageNet only with the class labels without any strong supervision so that the models may have a weaker localization capability than the models trained with more supervision, such as on detection and segmentation tasks. Second, our model incorporates much more deformable operations in the entire stages of a ResNet, so the dynamics of predicting offsets would be different from the deformable convolution where the only later stages have the deformable convolutions.

### E.3 PERFORMANCE TRADE-OFFS

This work presents the potential usefulness of parameter-free operations such as the max-pool as a building block rather than pushing the performance to the limit. We first visualize the ImageNet performance comparison with many efficient models shown in Table 3 in Fig.E.3. Fig.E.3 shows our models have competitive trade-offs between accuracy and speed. Since we do not modify ResNet50 but just replace the spatial operation into parameter-free operations such as the max-pool, our models do not have much benefits in the number of parameters and FLOPs.

Moreover, we visualize the ImageNet results including 1) top-1 accuracy on ImageNet; 2) mean Corruption Error (mCE) on ImageNet-C; 3) Area Under the precision-recall Curve (AUC) on ImageNet-O applying the efficient bottleneck into existing big CNNs, which is shown in Table 4, in Fig.E.4. Fig.E.4 shows our models achieve large improvements on the entire computational costs but show similar trade-offs of the baseline models in top-1 accuracy. This experiment originally aims to investigate the redundancy of using standard building blocks inside heavy CNN models. However, a simple replacement of the operations in existing models achieves significant efficiency; namely, in terms of the model speeds, the number of parameters, and FLOPs, our models reduce meaningful amounts without much accuracy loss. Furthermore, our models significantly outperform the baseline models with efficiency in mCE and AUC measures. Improving the ImageNet top-1 accuracy while having the advantages of the parameters-free operations will be our future work.

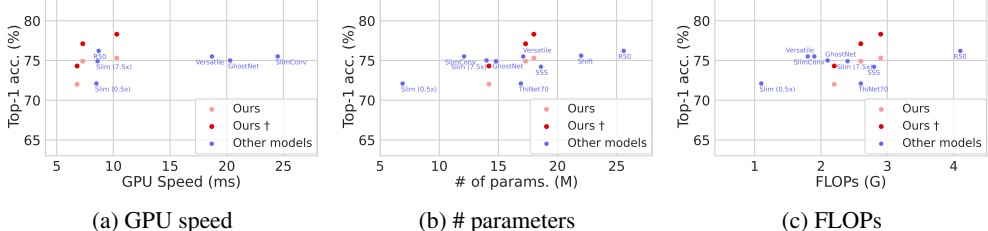

|     (a) GPU speed     |     (b) # parameters     |     (c) FLOPs     |
|---|---|---|

Figure E.3. **Performance Trade-offs of the models in Table 3.** We visualize the trade-offs between (a) accuracy and speed; (b) accuracy and # parameters; (c) accuracy and FLOPs, respectively. Ours denote the performance of Ours-R50 (max), Ours-R50 (*hybrid*), and Ours-R50 (deform_max); we also plot the models trained with the further training recipes.

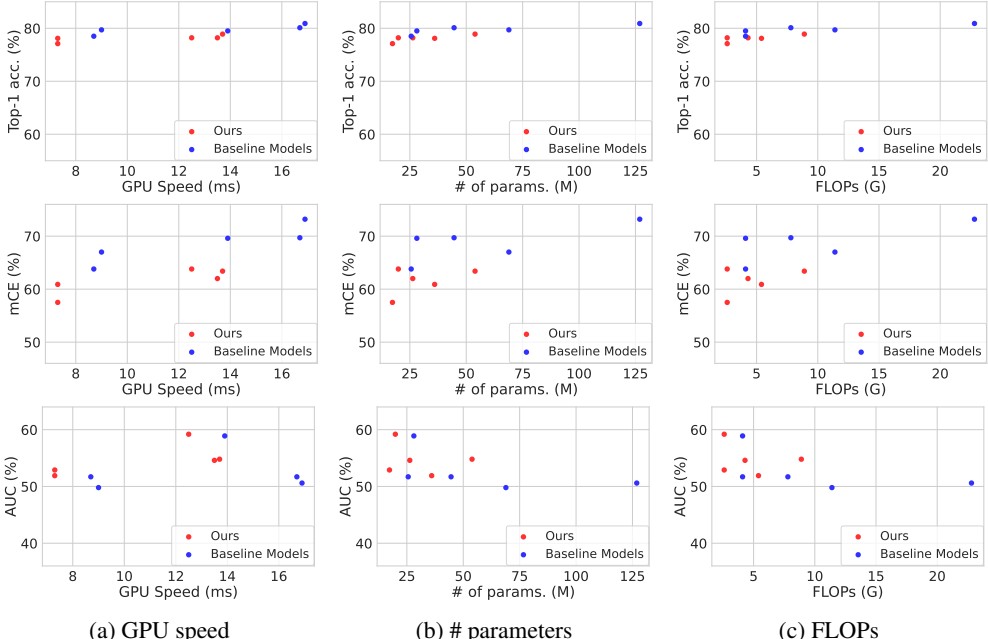

|     (a) GPU speed     |     (b) # parameters     |     (c) FLOPs     |
|---|---|---|

Figure E.4. **Performance Trade-offs of the models in Table 4.** We visualize the trade-offs between accuracy/error and (a) speed; (b) # parameters; (c) FLOPs, respectively. Each row employs 1) Top-1 accuracy on ImageNet; 2) mean Corruption Error (mCE) on ImageNet-C; 3) Area Under the precision-recall Curve (AUC) on ImageNet-O as the performance measure, respectively.

