# OpenReview forum: "Learning Features with Parameter-Free Layers"
_ICLR.cc/2022/Conference — ICLR 2022 Poster_

### Official Review · Reviewer_PJUA · 2021-10-26

**Correctness:** 3
**Technical Novelty And Significance:** 3
**Empirical Novelty And Significance:** 2
**Recommendation:** 6
**Confidence:** 4

**Main Review:**

Strength:
1. I think the idea of replacing the spatial operation (e.g., 3x3 conv) with some parameter-free/cheap operations is good. And the results showed that even simple max pooling could work well, which is inspiring.
2. The author conducted thorough ablation studies and evaluations on different architectures and datasets. These experiments clearly demonstrated the advantages and disadvantages of the method.
3. The paper is well written.

Weakness:
1. I think replacing the spatial operations with some cheap operations is a good idea, but the author could dive deeper into this. Currently, the author mainly conducted this with max pooling, and it achieves reasonable results but not surprisingly good. For example, in most cases, the inference speed is improved by 10%-20% but the accuracy is also decreased by 1-2 points. This seems to be a pretty normal accuracy-efficiency trade-off.
2. The author tried some improvements, i.e., deform_max, which I think is a good direction. But there are few experiments on it (only in Table 3). I am curious about how the deform_max works without improved training tricks and on other tasks.
3. The author claimed that depthwise conv is light in params/flops but heavy in inference speed (which I agree with), while the built-in max pooling is highly optimized and efficient. But in Table 4, the flops seems to be reduced by 30%+ while the inference time is only reduced by 10%-20%. Could the author explain this?
4. The author claimed the deform_max could be optimized further to speed up. I think the author could work on that and show how much speedup it has.


**Summary Of The Paper:**

This paper proposed to use some parameter-free operations to replace the spatial operations (e.g., 3x3 conv, 3x3 depth conv). The author shows that using the parameter-free operations (e.g., max pooling) can also achieve a good performance. The author conducted thorough ablations to show the effectiveness of parameter-free operations, and the method is evaluated on different architectures and datasets.

**Summary Of The Review:**

Overall, I think this paper proposed an inspiring idea but the author could explore further on this. The current results are not surprisingly good but promising. It would be good If the author could further improve the method.

=============== Post rebuttal ================

Thank the authors for the detailed response. I am satisfied with most of it. I still have some concerns about the proposed deform_max and its performance. The idea of replacing conv with pooling (or general parameter-free operations) seems promising. But the results in the paper are not suprising yet, even with the deform_max. The accuracy is about 1% lower than baseline, with 1.2 GFLOPs less computation but even more inference time. It would be good if the author could further improve and optimize their method. I would like to keep my score as weak accept.

---

> ### Author Response · Authors · 2021-11-16
> **Response to Reviewer PJUA (1/2)**
>
> We deeply appreciate the constructive review, and the valuable and insightful comments were very helpful to revise the paper. We have carefully read the comments and questions and tried to address all of them to be resolved through this response.
>
> **C1.** _I think replacing the spatial operations with some cheap operations is a good idea, but the author could dive deeper into this. Currently, the author mainly conducted this with max pooling, and it achieves reasonable results but not surprisingly good. For example, in most cases, the inference speed is improved by 10%-20% but the accuracy is also decreased by 1-2 points. This seems to be a pretty normal accuracy-efficiency trade-off._
>
> **A.** We agree that the performance improvements may not be surprising. In fact, we focused more on showing the potential usefulness of parameter-free operations such as the max-pool rather than pushing the performance to the limit with our method; as commented by Reviewer HgeU, this is the first attempt to investigate built-in parameter-free operations as a building block for network efficiency.
> Additionally, we would like to stress that in terms of the number of parameters and FLOPs, our models reduce meaningful amounts without much accuracy loss, as shown in Table 3 and Table 4. We believe this work may give insight into studying a new line of backbone using such parameter-free operations.
>
> **C2.** _The author tried some improvements, i.e., deform_max, which I think is a good direction. But there are few experiments on it (only in Table 3). I am curious about how the deform_max works without improved training tricks and on other tasks._
>
> **A.** We agree that one may be curious about the performance of the models with deform_max without improved training tricks. Therefore, we trained a ResNet50 with deform_max (the identical architecture of Ours-R50 (deform_max) in Table 3) with the basic 90-epochs training setting without any training tricks. Finally, we have achieved 75.3% top-1 accuracy (top-5: 92.5%), which shows the same accuracy trend with the models trained with the improved tricks. We have added the result in Table 3; here, we summarize the result below.
>
> |Models |Params. (M) | FLOPs (G) | GPU (ms)  |Top-1 (%) | Top-5 (%)|
> |:---|:--:|:-:|:-:|:-:|:-:|
> |R50 (max)             | 14.2 |2.2| 6.8 |72.0 |90.5|
> |R50 (hybrid)          | 17.3 |2.6 |7.3| 74.9| 92.2|
> |**R50 (deform max)** |18.0 |2.9 |10.3 |**75.3** |**92.5**|
>
> Furthermore, we verify the effectiveness of deform_max on other tasks; we chose the object detection task on the COCO dataset to compare with the results in Table 5. We have reported the AP scores of ResNet50 (deform_max) in Table 5; here, the results are summarized below:
>
> |Backbone | IN Acc. (%) | AP | GPU (ms) | Params. (M) | FLOPs (G)|
> |:---|:--:|:-:|:-:|:-:|:-:|
> |R50                       | **76.2** | 32.9 | 42.8 | 41.8 | 202.2|
> |R50 (Hybrid)          | 74.9 | 31.9 | 37.5 |33.6 |175.8|
> |**R50 (deform max)** | 75.3 | **33.2** | 47.7 |34.3 | 181.8|
>
> We observe that AP scores are significantly improved more than estimated by the ImageNet accuracy improvement. This indicates deform_max also works well on such an object detection task. We will perform more studies on the other tasks. It is worth noting that the AP score of the ResNet50 with deform_max on the COCO dataset has improved from 33.2 (reported score in Table 5) to 36.1 when trained with a new learning rate (0.01) and weight decay (1e-5).

---

> > ### Author Response · Authors · 2021-11-16
> > **Response to Reviewer PJUA (2/2)**
> >
> > **C3.** _The author claimed that depthwise conv is light in params/flops but heavy in inference speed (which I agree with), while the built-in max pooling is highly optimized and efficient. But in Table 4, the flops seems to be reduced by 30%+ while the inference time is only reduced by 10%-20%. Could the author explain this?_
> >
> > **A.** First, we want to make it clear that Table 4 contains the results replacing the regular convolution with the max-pool operation, not replacing the depthwise convolution in ResNets. Therefore, the FLOPs were significantly decreased over the speed improvements because the regular convolution is usually faster but has larger FLOPs than depthwise convolution.
> > Moreover, generally, an inference speed change is not likely to follow precisely proportional to a FLOPs change because speed is calculated by overall summation of memory access time and arithmetic calculation time that is only considered in computing FLOPs. The improvement of speed gets larger when compared with the depthwise convolution.
> >
> > **C4.** _The author claimed the deform_max could be optimized further to speed up. I think the author could work on that and show how much speedup it has._
> >
> > **A.** We agree that it would be better for the proposed deform_max operation to be optimized further as the max-pool does. However, because our implementation was based on a third-party implementation without expertise in CUDA-level optimization, it was hard to reach the standard optimization level to a built-in function such as the regular convolution in a deep learning library. We have a future goal to improve the operation faster based on an improved baseline code with further CUDA-level optimization.

---

### Official Review · Reviewer_KvAV · 2021-10-31

**Correctness:** 3
**Technical Novelty And Significance:** 2
**Empirical Novelty And Significance:** 3
**Recommendation:** 6
**Confidence:** 4

**Main Review:**

The framing of the method as more general parameter-free operation is over-stated for what has been demonstrated, which is limited to max-pooling (and average pooling in a small number of cases).  In particular, the abstract mentions "parameter-free" several times, and "max-pooling" not once.  (A second operation, deformable max-pool, is mentioned, but only as a direction for future work, and is not studied beyond one small experiment in the appendix).  Likewise, calling the new layers "efficient layers" seems a little inflated to me, if they are just max-pooling substitutions.  It seems more accurate to call them "max-pooling bottlenecks" or something specific to max-pooling.

Still, the empirical results are intriguing, showing that simple max-pooling ops can be used in place of convolutions much more than I might have expected.

Although I feel I was able to follow most of the paper well enough to understand the technique, I did feel like I was guessing a bit at several key points, and would have liked to have seen more precise definitions.  I have listed these in below detailed comments.

Overall, the empirical results are quite interesting, showing that max-pooling can be used in place of convs or depth-wise convs more extensively than I might have expected, and offers a meaningful datapoint in the wider context parameter-free operations.  While the most specific claims are supported by the experiments, I think the overall framing as a demonstration of parameter-free layers is over-generalized for what has been demonstrated around max-pooling.  There are also many details that I was unclear on, and feel could be more clearly described (below).


Additional comments/questions:


* A couple older works might also be interesting to discuss
  - Maxout Networks, Goodfellow et al 2013 https://arxiv.org/pdf/1302.4389.pdf --- uses max as a nonlinearity in place of relu
  - Inception v1, v2, v3, ... etc networks also use max pooling as a branch, and some of the cells found in the NAS in this paper look similar to Inception branches

* sec 3.2:  The use of W = s(g) doesn't quite correspond to what I think is the intended operation.  "W_v,:,: = s(g)" looks like there is one W matrix that is constructed globally based on the entire feature map g, and used used as a convolution kernel.  Instead, I think the intent is to describe a per-window operation, in which case there should be a dependency on the window location i,j.

* sec 3.2:  i,j are used twice in W, argmax description:  once in the definition (i,j)=argmax_hw, and once again in the expression in the argmax g_v,r*i+h,r*j+w.  most likely the i,j inside and outside the argmax are different (new names would help here).


* sec 3.3 multiple bottlenecks experiment:  I had trouble following the description of this experiment.  What was the exact architecture, and how were the replacements of convolutions performed?  How many layers were replaced at a time, and with which operations?  My best guess here, is that starting from a resnet with convolution residuals, new models were created by a procedure where (1) a set of one or more layers were selected for operation replacement (how?), and then (2) in these layers, 3x3 convolutions were replaced with either (a) max pooling or (b) depthwise convolution, creating pairs of models for cases (a) and (b).  The best 20% of models in each (a) and (b) conditions are then plotted in red and blue in Fig 2.  Is this correct?

* Fig 2:  It would also be interesting to see not just the best 20%, but also the other 80%, what the performance looks like as it degrades.


* sec 4.1:  This section is the first use of the phrase "efficient bottleneck" which is a little confusing, as it would seem the same module was investigated in the cifar10 experiments as well, and described in sec 3.  I think it would be clearer to define the "efficient bottleneck" in sec 3 using an equation that closely parallels eqns (1)..(4).  I believe that the "efficient bottleneck" operation is the eq (2) but replacing the innermost double-sum (over space window and \roh*c channels) with per-channel max pooling over k sized windows.


* transformer and Fig A.1:  I'm not sure exactly where the maxpooling operation is applied.  In fig A.1(b), there are three paths into the attention layer, which seem to indicate these correspond to q,k,v.  But the same three paths go into the eff-layer in A.1(d) --- does this layer actually combine these three sets of features somehow, or is it applied to just one set?


* "deformable max-pool operation":  how are the predicted locations are found for this operation, are there parameters involved in this step?  what parts do or do not involve parameters?


* Tables 3,4:  It would be informative to plot the first three columns of tables 3 and 4 vs top1 in a set of scatterplots, to better see the speed/performance tradeoff and whether this method moves along the frontier or pushes it.

* Fig 1:  would be nice to have the condition names in left/top of the plot grid, not just the caption



**Summary Of The Paper:**

This paper argues for more extensive use of max-pooling layers in image classification networks, as an inexpensive substitute for convolutions.  Recent works have removed parameters (and often-expensive operations) using bottlenecks, depthwise/channel-wise operations, etc., and this paper pushes further in this direction by studying max-pooling as a cheap and effective spatial combination mechanism in the context of recent resnet-derived and image transformer architectures.  Empirical experiments are performed on CIFAR-10 and three Imagenet variants, demonstrating the effectiveness of incorporating max-pooling layers beyond applications in strided layers and secondary branches.


**Summary Of The Review:**

Overall, the empirical results are quite interesting, showing that max-pooling can be used in place of convs or depth-wise convs more extensively than I might have expected, and offers a meaningful datapoint in the wider context parameter-free operations.  While the most specific claims are supported by the experiments, I think the overall framing as a demonstration of parameter-free layers is over-generalized for what has been demonstrated around max-pooling.  There are also many details that I was unclear on, and feel could be more clearly described.

---

> ### Author Response · Authors · 2021-11-16
> **Response to Reviewer KvAV (1/3)**
>
> We deeply appreciate the detailed and constructive review. The valuable and insightful comments and advice were very helpful to revise the paper. We have carefully read the comments and questions and tried to address all of them to be resolved through this response.
>
> **C1.** _About using “parameter-free operations” as a general term:_
>
> **A.** Since we focused on the ‘‘parameter-free” aspect of particular operations and considered the extensibility of the work in mind, we used the term parameter-free operation multiple times in our paper. As pointed out, this paper mainly argues parameter-free operations for the max-pool and the avg-pool operations; this was to investigate the potential of using such simple built-in functions in a deep learning library to reach the performance trade-off of a general model such as ResNet.
>
> In fact, we mentioned in the Introduction section that we revisit the max-pool and average-pool operations for our study from the beginning (please refer to the statement at the bottom of p.1 starting with the sentence “we revisit the popular parameter-free operations, the max-pool and the avg-pool operations ~”). However, to alleviate the misunderstanding, we have added the arguments in the Abstract and Introduction section to clarify that “we are mainly handling the max-pool and the avg-pool operations here.” We have further similarly refined the other parts of the paper as possible as we can. Please refer to the revised part highlighted in red in the paper.
>
> **C2.** _Suggestion for “Maxout Networks” and “Inception v1,v2,v3 ..” papers to be discussed:_
>
> **A.** Thank you for suggesting the interesting literature to discuss. The Maxout paper gave us an intuition as to why the max-pool operation works. The Inception papers are another empirical backup for using the max-pool operation in the main building block. We have discussed Maxout in Appendix D and cited the Inception papers at the end of the paragraph in Section 3.3.
>
> **C3.** _About fixing the notations (W_v,:,: = s(g)  and  (i,j)=argmax_hw)  in sec 3.2_
>
> **A.** We are sorry for the confusion. We have refined the notation W_v,:,: = s(g) by W_v,h,w = s(g_v,r*i+h, r*j+w)y. We have also redefined the notation by following the suggestion distinguishing two i,j sets by changing one to h* and w*. Please refer to the revised notations in the paper.
>
> **C4.** _sec 3.3 multiple bottlenecks experiment: I had trouble following the description of this experiment. What was the exact architecture, and how were the replacements of convolutions performed? How many layers were replaced at a time, and with which operations? My best guess here, is that starting from a resnet with convolution residuals, new models were created by a procedure where (1) a set of one or more layers were selected for operation replacement (how?), and then (2) in these layers, 3x3 convolutions were replaced with either (a) max pooling or (b) depthwise convolution, creating pairs of models for cases (a) and (b). The best 20% of models in each (a) and (b) conditions are then plotted in red and blue in Fig 2. Is this correct?_
>
> **A.** The guess is right. We used the standard ResNet with eight bottlenecks inside (i.e., ResNet-26). Either replacement with the max-pool and depthwise convolution from the regular convolution was done exhaustively in ResNet-26, so we have 2^8=256 models to figure out the performance trade-off. After that, Fig.2 was plotted by picking the best 20% trade-off models for better visibility. This was, in fact, stated in the original submission, but we are sorry for making it difficult to understand. We have made the text clearer; please see the highlighted text in red in Section 3.3.
>
>
> **C5.** _Fig 2: It would also be interesting to see not just the best 20%, but also the other 80%, what the performance looks like as it degrades._
>
> **A.** We have plotted the performance of the entire models trained in the study of multiple bottlenecks.  Please refer to the plots in Fig.E.1. The figure includes the 20% best-performing models plotted in Fig.2. As expected, the models using many max-pool operations have degraded accuracies but show clear speed benefits; there is about a 1.5% accuracy gap between the fastest models using each operation yet more than a 5ms speed gap between them.

---

> > ### Author Response · Authors · 2021-11-16
> > **Response to Reviewer KvAV (2/3)**
> >
> > **C6.** _sec 4.1: This section is the first use of the phrase "efficient bottleneck" which is a little confusing, as it would seem the same module was investigated in the cifar10 experiments as well, and described in sec 3. I think it would be clearer to define the "efficient bottleneck" in sec 3 using an equation that closely parallels eqns (1)..(4). I believe that the "efficient bottleneck" operation is the eq (2) but replacing the innermost double-sum (over space window and \roh*c channels) with per-channel max pooling over k sized windows._
> >
> > **A.** The efficient bottleneck was defined in Section 4.1 but is the same bottleneck used in Section 3, as pointed out. Following the suggestion, we have reordered the texts and defined the efficient bottleneck in Section 3 for clarity.
> >
> > **C7.** _transformer and Fig A.1: I'm not sure exactly where the maxpooling operation is applied. In fig A.1(b), there are three paths into the attention layer, which seem to indicate these correspond to q,k,v. But the same three paths go into the eff-layer in A.1(d) --- does this layer actually combine these three sets of features somehow, or is it applied to just one set?_
> >
> > **A.** Yes, we regard the three paths as a single path by computing tripled-channel output by the linear projection of the input; this is identical to the concatenation in the channel direction of the projected q,k, and v. Then, the output is fed into the spatial max-pool operation after reshaping it to 2d-features for each channel (i.e., reshaping from a 3d-tensor to a 4d tensor). Then, the tripled channels are reduced by the existing projection layer after self-attention. We have revised Section 4.2 for clarity.
> > We did not change the fundamental design of three paths (q,k, and v) to modify the vision transformers (ViT and PiT both) to verify a simple, practical use case of the max-pool operation inside the transformer block.
> >
> > **C8.** _"deformable max-pool operation": how are the predicted locations found for this operation, are there parameters involved in this step? what parts do or do not involve parameters?_
> >
> > **A.** It is similar to the architectural implementation in the deformable convolution. We replace the 3x3 convolution in the deformable convolution with the max-pool operation while remaining other convolution layers that compute offsets unchanged. Therefore the location predictor has weight parameters to predict the locations. A future direction can be to vanish those parameters. We have clarified this in the deformable max-operation paragraph in Section 5.1.
> >
> > Predicted locations: Following the suggestion, we visualize the predicted locations (i.e., offsets) from query inputs randomly picked from the ImageNet validation set. Please refer to Fig.E.2. We visualize the aggregated offsets from the last two deformable max-pool operations. Ours show promising results that localize either the foreground or background compared with the regular convolution illustrated by fixed points.
> >
> > We would like to stress that the models trained only with the classification have a weaker localization capability compared with the models trained with more supervision such as on detection and segmentation tasks. Furthermore, our model involves the deformable max-operation in the entire stages of a network, so the dynamic of predicting offsets would be different from the output of the deformable convolution where the only later stages have the deformable convolutions.
> >
> > **C9.** _It would be informative to plot the first three columns of tables 3 and 4 vs top1 in a set of scatterplots, to better see the speed/performance tradeoff and whether this method moves along the frontier or pushes it._
> >
> > **A.** We focused more on showing the potential usefulness of parameter-free operations such as the max-pool rather than pushing the performance to the limit with our method. As commented by Reviewer HgeU, this is the first attempt to investigate built-in parameter-free operations as a building block for network efficiency. We believe this work may give insight into studying a new line of backbone using such parameter-free operations.
> >
> > We have drawn three plots with the measures in the three columns in Table 3 and Table 4, respectively; please see Fig.E.3 and Fig.E.4. We observe that Fig.E.3 shows a better trade-off between accuracy and speed, so ours seems to push forward the trade-off frontier. However, Fig.E.4 shows our models show similar trade-offs of the baseline models, but it is clear that a simple replacement in existing models could achieve significant efficiency. Namely, our models reduce meaningful amounts without much accuracy loss in terms of the model speeds, the number of parameters, and FLOPs. Our further work will focus more on improving the accuracy while maintaining the advantages of the parameters-free operations.

---

> > > ### Author Response · Authors · 2021-11-16
> > > **Response to Reviewer KvAV (3/3)**
> > >
> > > **C10.** _Fig 1: would be nice to have the condition names in left/top of the plot grid, not just the caption_
> > >
> > > **A.** We revised the condition names at the left and the top of the plot. Please check the revised Fig.1.

---

> ### Author Response · Authors · 2021-11-29
> **A gentle reminder: please let us know if you have additional questions**
>
> Dear Reviewer KvAV,
>
> We appreciate your time to review and the constructive comments to encourage our work. We want to leave a gentle reminder due to nearing the end of the discussion period. We have tried to address all your concerns by providing more explanations and results. Please go over our response, and if you have additional questions, please let us know.
>
> Thank you,
>
> Authors.

---

> ### Comment · Reviewer_KvAV · 2021-11-29
> **post-rebuttal**
>
> Thanks for the responses.  I've raised my score to 6.  The edits to the abstract and intro make it much clearer at the  start that the study is limited to max-pooling, and I think sufficiently aligns the claims to the results.  The additional edits to smaller points clear up many of the details I was unsure about as well, including the new Fig E.1, which is more complete, and the new interpretation in D.1 is interesting as well.

---

### Official Review · Reviewer_DuSH · 2021-11-01

**Correctness:** 3
**Technical Novelty And Significance:** 3
**Empirical Novelty And Significance:** 2
**Recommendation:** 6
**Confidence:** 4

**Main Review:**

Pros:
+ A simple yet effective finding of using parameter-free operations to achieve speedup with reasonable performance drop.
+ Very extensive experiments to support authors' claim on the effectiveness of the parameter-free operations.

Cons:
- In spite of extensive experiments to demonstrate the effectiveness, it lacks some insightful explanation on the reason why it works, why it could achieve a good balance between speed and performance.
- In the experiment of hybrid architecture with efficient bottlenecks, what about putting the designed efficient block at different position of the architecture? What about alternatively stack baseline and efficient blocks?
- The performance drops more 1% in AP while reducing less than 5 ms running time in the object detection task. It seems not to be a good tradeoff between performance and speed. Any explanation on such performance in the object detection task? Any solution to address this issue?
- Why does ViT-S with maxpooling shows slight improvement in performance while drop in speed?

**Summary Of The Paper:**

In this paper authors propose to use simple parameter-free operations to replace several kinds of trainable layers to achieve speedup on hardware with less performance drop. Extensive experiments have been done to demonstrate the effectiveness of the proposed method, including experiments on replacement in both single and multiple bottlenecks and neural architecture search. In addition, authors also employ such parameter-free operations to redesign CNNs and vision transformers for a good balance between speed and performance.

**Summary Of The Review:**

This paper shows a simple yet effective finding and validate this with extensive experiments. However, it lacks some insightful explanation and novelty.

---

> ### Author Response · Authors · 2021-11-16
> **Response to Reviewer DuSH (1/2)**
>
> We deeply appreciate the constructive review with the valuable and insightful comments. All the comments were very helpful to revise the paper. We have carefully read the comments and tried to address all the concerns to be resolved through this response.
>
> **C1.** _In spite of extensive experiments to demonstrate the effectiveness, it lacks some insightful explanation on the reason why it works, why it could achieve a good balance between speed and performance._
>
> **A.** To understand why the max-pool operation inside a bottleneck works well, we linked with the previous literature Maxout (the idea came up with the comment by reviewer PJUA). Maxout performs after linear layers or convolutions to ensemble the outputs in a nonlinear manner, which causes the improvement of model capacity. We believe there is a relationship between the Maxout operation and the max-pool operation inside the efficient bottleneck. Please refer to Appendix D.1.
>
> Since the spatial parameter-free operations such as the spatial max-pool have a receptive field like convolutions, we conjecture that such simple operations may capture striking pixels (or patches). We further conjecture that a 1x1 convolution can complement the expressiveness of a parameter-free operation such as the max-pool operation by nonlinearly mixing the features. To make a ground for the conjectures, we visualized the intermediate features of successive layer outputs. Please refer to Appendix D.2 with Fig.D.1 and Fig.D.2.
>
> The good trade-offs between accuracy and speed shown in the experiments may stem from the parameter-free operations’ speed advantage over convolutions. Specifically, there is no arithmetic computation time for the parameter-free operations such as the max-pool, only the time it takes to access memory. Moreover, the efficiency gets significant with a built-in optimized function provided in a deep learning library. A strict and theoretical coverage of the effectiveness of parameter-free operations will be our future work.
>
> **C2.** _In the experiment of hybrid architecture with efficient bottlenecks, what about putting the designed efficient block at different positions of the architecture? What about alternatively stack baseline and efficient blocks?_
>
> **A.** Studying with as many candidates in which the efficient blocks and bottlenecks are placed in different positions is essential to understand hybrid architecture concretely. There are many candidate models with the designed efficient bottlenecks at various positions; for example, we have overall 2^16 models designed in a hybrid manner based on ResNet50, where the number of blocks is 16. Because it was hard to train all the models exhaustively to compare with each other, we alternatively studied at the level of ResNet’s stage as illustrated in Section 4.1. We are planning to train more models and validate their performance.
>
> New model - Thank you for suggesting a way of designing a model with the efficient bottleneck. Following the suggestion, we have trained the model by stacking the baseline bottleneck and the efficient block alternately in ResNet50.  We trained the models on ImageNet with the 90-epochs training setting used in the paper. Here, we provide the ImageNet performance of the new architecture (we call it E/B): 75.3% with 7.7ms latency, which is another promising result compared with the competitors in Table 2. The architecture relatively has many standard bottlenecks, so the number of parameters (21.1M) and FLOPs (3.2B) are bigger. Please refer to the summarized result:
>
> |Models |GPU Lat. (ms) |Top-1 acc.(%)|
> |:---|:--:|:-:|
> |Hybrid |7.3 | 74.9|
> |**E / B** | 7.7 | **75.3** |
>
>
> We have added the result of the new model in Table 2 and revised the text (highlighted in red) in Section 4.1.

---

> > ### Author Response · Authors · 2021-11-16
> > **Response to Reviewer DuSH (2/2)**
> >
> > **C3.** _The performance drops more 1% in AP while reducing less than 5 ms running time in the object detection task. It seems not to be a good tradeoff between performance and speed. Any explanation on such performance in the object detection task? Any solution to address this issue?_
> >
> > **A.** This experiment aims to validate whether the proposed pretrained-backbone (many parameter-free operations are inside) can be transferred to other tasks without hard engineering. To this end, we chose the object detection task and verified whether the ImageNet performance is still the basis for transfer learning performance with a fundamental object detection architecture.
> >
> > Therefore, we did not refine the network architecture and train object detection models under multiple setups. We trained once with the identical Faster RCNN under a single fixed training setup, which is the standard setup for the ResNet backbone in Faster RCNN. As a result, we observe the 1% AP gap on COCO detection from 1.3% ImageNet accuracy, which seems to follow a general trend of transfer learning accuracy when using different ImageNet-pretrained backbones as shown in previous literature.
> >
> > Solutions? - We are considering some ways of raising AP scores. We may renovate the detection head for parameter-free operations or search for new training setups for the particular backbone where massive parameter-free layers are inside. Among the options, we have tried the second option and achieved the AP improvement from 31.9% to 32.9% by changing the learning rate from 0.005 to 0.01 and the weight decay from 1e-4 to 1e-5; since we could not explore the hyper-parameter space exhaustively, there may be room to improve the score further.
> >
> > Furthermore, we added the COCO detection result with the ImageNet-pretrained ResNet50 (deform_max) in Table 2. It is worth noting that the reported AP score of 33.2 is still better than the baseline (32.9), but we found it increases to 36.1% when using the above new setting.
> >
> > **C4.** _Why does ViT-S with maxpooling shows slight improvement in performance while drop in speed?_
> >
> > **A.** First of all, as stated in Appendix A (p.14), we intended not to modify the vision transformers (ViT and PiT both) as much to verify a practical and straightforward use-case of the max-pool operation. Therefore, directly incorporating the max-pool operation into the transformer may not be efficient even though the max-pool is efficient; we believe there is a potential to design a ViT which is more friendly to max-pool-like operations.
> >
> > Specifically, the spatial max-pool/avg-pool in many deep learning libraries should be fed with the input of 4d tensors (e.g., B x C x H x W in PyTorch). Therefore, intermediate 3d-tensor features in a ViT need to be reshaped into 4d tensors incurring the speed to be slowed down. Note that this has partially resolved in PiT, which processes some intermediate 4d features resulting in the improved throughput (+48), as shown in Table 6.
> >
> > For the accuracy improvements, ViTs have been known to have low generalization ability with insufficient data (as shown in Table 6), shown in many vision transformer papers, including ViT and PiT. We conjecture the max-pool operation performs like a convolution with inductive bias, so the accuracy may be improved when using relatively little data.

---

> ### Author Response · Authors · 2021-11-29
> **A gentle reminder: please let us know if you have additional questions**
>
> Dear Reviewer DuSH,
>
> We appreciate your time to review and the constructive comments to encourage our work. We want to leave a gentle reminder due to nearing the end of the discussion period. We have tried to address all your concerns by providing more explanations and results. Please go over our response, and if you have additional questions, please let us know.
>
> Thank you,
>
> Authors.

---

### Official Review · Reviewer_HgeU · 2021-11-02

**Correctness:** 4
**Technical Novelty And Significance:** 4
**Empirical Novelty And Significance:** 4
**Recommendation:** 8
**Confidence:** 4

**Main Review:**

Strengths:
1. Simple but novel idea. The first attempt to investigate built-in parameter-free layer as a building block for network efficiency.
2. Good and convincing empirical studies. The proposed method achieves compelling results on ImageNet dataset (Table 3, best trade-off for GPU), with detailed analysis on different network variants (CNN and ViT, + deformable operations etc).
3. Clear and sound technical presentation. The paper clearly elaborates the technical design and insight, and illustrates the difference with existing efficient building block.
4. Interesting and inspiring findings. For example, similar performances have been achieved using max-pool and avg-pool operation (Appedix B).

Questions:
1. Is there any other parameter-free operations that could possibly achieve good trade-off except max and avg operation?


**Summary Of The Paper:**

This paper focuses on designing efficient deep networks with a proposed novel parameter-free layers. Drawing the spirit of depth-wise convolution, shift operation, the paper studies the ways of inserting parameter-free operations into the building block using NAS technique. The resulted models perform both effectively and efficiently on ImageNet dataset.

**Summary Of The Review:**

A good paper. I definitely suggest acceptance.

---

> ### Author Response · Authors · 2021-11-16
> **Response to Reviewer HgeU**
>
> We deeply appreciate the positive reviews along with valuable and insightful comments to encourage our work. We have carefully read the comment and addressed the concern through this response.
>
> **C1.** _Is there any other parameter-free operations that could possibly achieve good trade-off except max and avg operation?_
>
> **A.** Thank you for suggesting an idea that can be a new direction for further study. Popular pooling operations such as the sum-pool or variance-pool may be other options to replace the max-pool and avg-pool operations. We found LPPool2d that contains both the sum and the variance operations by adjusting the hyper-parameter p (please refer to the description in  https://pytorch.org/docs/stable/generated/torch.nn.LPPool2d.html#torch.nn.LPPool2d). Thanks to the PyTorch library, which provides the built-in function, it seems that such an operation would show a good trade-off as well. We will explore more parameter-free operations and thoroughly study them, including the LPPool operation.

---

### Author Response · Authors · 2021-11-16
**General Response**

We thank all the reviewers for the thorough and constructive reviews with valuable and insightful advice. We appreciate the positive comments from all reviewers, which are 1) simple but novel idea, good and convincing empirical studies, clear and sound technical presentation, and interesting/inspiring findings by **Reviewer HgeU**; 2) a simple yet effective finding and very extensive experiments by **Reviewer DuSH**; 3) intriguing and interesting empirical results by **Reviewer KvAV**; 4) good idea with inspiring results, thorough ablation studies and evaluations with clear demonstration, and well-written paper by **Reviewer PJUA**.

We tried to address all the comments to revise the paper by following the reviewers’ comments. We newly wrote **Appendix D** and **Appendix E** to reflect all the comments. Throughout the revised paper, we highlighted the newly-added or edited materials **in red**. Please refer to the following summarization of the materials as follows:
- Added the discussion of understanding why such a parameter-free operation works well in Appendix D (**C1** by **Reviewer DuSH**)
- Added feature visualizations of the images from the ImageNet validation set in Fig.D.1 and Fig.D.2 for **C1** by **Reviewer DuSH**
- Added the ImageNet result of the newly designed model using bottleneck and the efficient bottleneck with alternately in Table 2 (**C2** by **Reviewer DuSH**)
- Clarified using the term parameter-free operation by specifying the use of the max-pool and avg-pool operations in the Abstract and the Introduction section (**C1** by **Reviewer KvAV**)
- Discussed “Maxout Networks” in Appendix D.1 and “Inception v1,v2,v3 ..” papers in Section 3.3 (**C2** by **Reviewer KvAV**)
- Clarified the terms and rearranged/refined texts (**C3**, **C4**, **C6**, **C7**, and **C10** by **Reviewer KvAV**)
- Added an extended plot of Fig.2 in Fig.E.1 (**C5** by **Reviewer KvAV**)
- Added visualization of predicted offsets by the deformable max-pool operation in Fig.E.2 (**C8** by **Reviewer KvAV**)
- Added two performance trade-off plots of Table 3 and Table 4 in Fig.E.3 and Fig.E.4, respectively (**C9** by **Reviewer KvAV**)
- Added the ImageNet result of ResNet50 (deform_max) in Table 3 (**C2** by **Reviewer PJUA**)
- Added the COCO detection result of ResNet50 (deform_max) in Table 5 (**C2** by **Reviewer PJUA**)

Detailed responses to other comments are left in individual comments.

---

### Decision · Program_Chairs · 2022-01-20

**Decision:**

Accept (Poster)

**Comment:**

This paper analyzes the extent to which parameterized layers within a CNN can be replaced by parameter-free layers, with specific focus on utilizing max-pooling as a building block.  After the author response and discussion, all reviewers favor accepting the paper.  The AC agrees that its empirical results open a potentially interesting discussion on network design.